# The Current Developments in Medicinal Plant Genomics Enabled the Diversification of Secondary Metabolites’ Biosynthesis

**DOI:** 10.3390/ijms232415932

**Published:** 2022-12-14

**Authors:** Mohammad Murtaza Alami, Zhen Ouyang, Yipeng Zhang, Shaohua Shu, Guozheng Yang, Zhinan Mei, Xuekui Wang

**Affiliations:** College of Plant Science and Technology, Huazhong Agricultural University, Wuhan 430070, China

**Keywords:** medicinal plants, genomics, multi-omics, secondary metabolites, biosynthetic pathways

## Abstract

Medicinal plants produce important substrates for their adaptation and defenses against environmental factors and, at the same time, are used for traditional medicine and industrial additives. Plants have relatively little in the way of secondary metabolites via biosynthesis. Recently, the whole-genome sequencing of medicinal plants and the identification of secondary metabolite production were revolutionized by the rapid development and cheap cost of sequencing technology. Advances in functional genomics, such as transcriptomics, proteomics, and metabolomics, pave the way for discoveries in secondary metabolites and related key genes. The multi-omics approaches can offer tremendous insight into the variety, distribution, and development of biosynthetic gene clusters (BGCs). Although many reviews have reported on the plant and medicinal plant genome, chemistry, and pharmacology, there is no review giving a comprehensive report about the medicinal plant genome and multi-omics approaches to study the biosynthesis pathway of secondary metabolites. Here, we introduce the medicinal plant genome and the application of multi-omics tools for identifying genes related to the biosynthesis pathway of secondary metabolites. Moreover, we explore comparative genomics and polyploidy for gene family analysis in medicinal plants. This study promotes medicinal plant genomics, which contributes to the biosynthesis and screening of plant substrates and plant-based drugs and prompts the research efficiency of traditional medicine.

## 1. Introduction

Medicinal plants produce thousands of secondary metabolites that have been used in traditional medicine for centuries and provide essential elements for industries, such as drugs, condiments, food additives, and essential oils. However, secondary metabolites synthesis in a low concentration in plants, the genetics of biosynthetic pathways, and the regulation of these pathways must be studied to overcome the low synthesis of various secondary metabolites in plant cells. The high-quality genome sequencing paired with metabolomics and proteomics increased our understanding of biosynthesis pathways toward expanding production of metabolites.

The progress of technology has played a crucial role in the development of plant genomics. Every technological innovation has dramatically promoted the advancement of genomics. Since the first-generation sequencing technology was invented in the 1970s, in just 50 years, sequencing technology has experienced three generations of innovation (Appendix A). It is still booming and changing with each passing day [1]. The development of genome sequencing, assembly technologies, and supporting bioinformatics analysis tools has injected additional vitality into the comprehensive development of medicinal plant genomics and molecular biology, making it possible to obtain high-quality medicinal plant genomes. A high-quality genome sequence will help to carry out extensive and in-depth molecular genetics research and play a significant role in identifying genes related to the biosynthesis of secondary metabolites in medicinal plants and clarifying the historical evolution status. The completion of whole-genome sequencing has laid a solid foundation for carrying out fundamental research on medicinal plants’ molecular breeding and accelerating the cultivation process of varieties. In addition, by employing the genetic data and regulatory network of the species and the omics technologies in accordance, the genomics research of medicinal plants aims to expose their influence on the human body from the level of the genome and understand their molecular mechanism to prevent human illnesses. Recently, most newly published medicinal plant genomes have used third-generation sequencing technology combined with second-generation short reads and high-throughput chromosome conformation capture (Hi-C) sequencing, used, for example, with *Coptis chinensis* Franch [2], *Tripterygium wilfordii* [3], *Sarcophaga peregrina* [4], *Salvia miltiorrhiza* Bunge [5], *Aquilegia oxysepala* var. kansuensis [6], *Ophiorrhiza pumila* [7], tea [8], and many others, as shown in Appendix A.

The acquisition of genome sequence facilitates comparative genomics study and provides valuable resources for analyzing secondary plant metabolites and their synthesis; it has brought new insights into the study of medicinal plants’ diversity and evolution. Previous reviews have reported on the sequencing of medicinal plants’ genomes, such as Cheng et al. (2021) [9], who reviewed the plant genome situation, sequencing technology development, and medicinal plant genome application. Hamilton and Robin Buell (2012) [10] summarized the status of the sequenced plant genome, Chen et al. (2018) [11] the angiosperm plant genome, and Jiao and Schneeberger (2017) [12] the impact of long-read genome sequencing (third-generation sequencing) technology on plants’ genomes. In recent years, the reduced cost and time enabled the sequence of the medicinal plant genome and found the biosynthesis pathways of many secondary metabolites. Yet, the sequencing of medicinal plants’ genomes is far behind that of microorganisms, crops, and animals; the applications of high-quality genome assembly to find the biosynthesis pathways of secondary metabolites and related key genes still have not been fully elucidated, and the studies to uncover them need to be strengthened.

In this review, we conduct a systematic review of the medicinal plant genome and evolution. Moreover, we explore the development of multi-omics (genomics, transcriptomics, and metabolomics) approaches for discovering key genes related to secondary metabolite biosynthesis in medicinal plants. This study could be foundational information for biomedicine and gene identification and significantly accelerate the discovery of genes and metabolic engineering.

## 2. Comparative Genomics and Evolutionary Analysis Allowed Us to Identify the Genes in Secondary Metabolites’ Biosynthesis

An important subfield of genomics is comparative genomics. Biological research contrasts the structural and genomic properties of the genomes to comprehend the genetic linkages, mechanisms of expression, and functions of various species [13]. Similarity and homology are the basic principles of comparative genomics [14]. Similarity only obtains the degree of similarities and differences between different species through comparing simple nucleotide or amino acid sequences and does not involve evolutionary origin, kinship, and the relationship between structure and function. Homology is the exact evolutionary origin of different species, which shows the identity of all or functionally important conserved domains of nucleotide or amino acid sequences. The essence of similarity is the quantitative relationship of similarity between genome sequences of different species, and homology is the evolutionary common-source relationship of genes among species inferred from additional biological data and statistical analysis. Homology can be further divided into orthology and paralogy [15]. Corresponding linear homologous genes refer to homologous genes distributed in the genomes of different species and derived from speciation events, originated from an ancestor in an evolutionary relationship, and inherited vertically. Although there may be variation among homologous genes during species differentiation, they are highly conservative in structure and function and maintain the similar functions of ancestral genes, and based on such characteristics, the phylogenetic tree constructed by lineal homologous genes can reveal the evolutionary relationship between species. Collateral homologous genes are formed by lateral transmission due to the doubling of ancestral genes in the same species. They belong to the same gene. After continuous replication, the genes formed by replication enter similar but different evolutionary pathways due to mutation. Therefore, new gene functions are formed in the process of continuous variation and adaptation.

Based on the genetic link between the studied species, comparative genomics may be further subdivided into interspecific and intraspecific comparative genomics [16]. The typical species of comparative genomics among medicinal plants are *C. chinensis* and four additional species of the Ranunculales; Ranunculales had 42 gene families and 352 genes that were unique to the order. Functional annotation revealed that these gene family members were engaged in the metabolism of bisbenzylisoquinoline (BIA) alkaloids, suggesting that they could contribute to the Ranunculales’ production of common precursors for the subsequent biosynthesis of other BIA alkaloids [2]. Additionally, comparative genomic analysis between *Scutellaria aicalensis* and *Scutellaria barbata* revealed that the current long terminal repeat (LTR) might result in chromosomal extension and rearrangement. Tandem duplication of paralogs after their speciation may have led to the varied development of flavonoid biosynthesis gene families, which provided a crucial starting point for research on the evolution and chemo-diversity of the Lamiaceae [17]. Numerous research has shown how comparative genomics plays a key role in exposing the traits of complex genomes (polyploidy, high heterozygosity, and high duplication) as well as the genesis and evolution of species [18,19,20]. With the launch of genome sequencing projects for more and more species, such as the 5000 insect genome (i5k) project [21], 10,000 vertebrate genome (g10kcos) project [22], 10,000 plant genome project (10 kp) [23], 10,000 bird genome (b10k) project [24], and earth biological genome project (EBP) [25], and the release of the high-quality genome, it is possible to study comparative genomics across more species and even the whole-order species. We can explore the evolutionary relationship of species through a more macro perspective, find the source of life explosion in the process of earth evolution, and connect the bridge between genes and phenotypes and products.

We can now fully use comparative and phylogenetic techniques for gene identification due to the increasing number of genomic sequences that are now accessible. The evolutionary links of critical metabolic enzymes were previously studied using single genes, and only a small quantity of sequencing data from a few species was employed. As a consequence of advancements in next generation sequencing (NGS) technology, more detailed examinations into the mechanisms by which metabolic pathways developed are now possible for primitive angiosperm taxa such as Amborella, gymnosperms, lycopods, and mosses as well as many species within a taxonomic family [26]. Owing to the availability of many genomes from the same or closely related taxonomic groups, it is now possible to identify orthologous across species and taxa in a biosynthetic pathway and forecast the appearance of a feature due to synteny (same gene order), which shows common evolution [27]. Due to a large number of genome sequences within particular clades and the variety of genome sequences throughout the plant kingdom, comparative analysis across and within species may be a potent tool for discovering the evolutionary origins of biochemical processes.

## 3. Genome Duplication Contributed to the Biosynthesis of Secondary Metabolites in Medicinal Plants

The genetic material is doubled after whole genome duplication (WGD) or polyploidization of the plant genome, and the duplicated genes are differentiated in distinct ways. This process might result in the plant genome producing novel gene functions, metabolic pathways, and phenotypic traits. The ability to identify the genes involved when combined with comparative genome and gene family analysis techniques may increase breeding options while preserving species variety. The differentiation of gene function brought on by polyploidy may aid in developing novel species-specific features. According to studies, the production of several essential elements is strongly tied to genomic polyploidy events in the Cruciferae, Gramineae, Leguminosae, Compositae, Solanaceae, and angiosperms [28]. The most concrete proof is that many essential features are genetically regulated by a multi-copy gene regulatory network created by the genome polyploidy event. Close to the time of the family-specific genome polyploidy event is when a species or trait becomes unique. For instance, tea genome sequencing indicates that there have been two WGD events, the most recent of which took place around 30–40 million years ago (MYA), giving rise to catechins and caffeine, the key secondary metabolites responsible for the taste of tea (the primary bitter substance) [29].

Tandem duplication has traditionally been considered a required method by which plants increase their accumulation of secondary metabolites. For instance, the biosynthetic pathways for morphine [30] and caffeine [31] in poppy and tea plants underwent significant tandem gene duplication events that expanded their respective gene networks. Studies have also shown that WGD contributes to the manufacture of secondary metabolites [32], which is another vital element in the development of stress tolerance in plants [33,34]. Plant genomes are often complicated due to cyclical polyploidization and genome rearrangement, making it challenging to comprehend how they formed and evolved and discover novel gene functions [35].

Genes involved in biosynthesis had a significant increase in copy number. After conducting comparative genomic research with 10 sample plants, it was discovered that the tea genome considerably increased the copy number of genes synthesizing terpenes and other compounds associated with tea taste. It resulted in a more pronounced increase in tea’s scent [36]. As a result, occurrences of genomic polyploidy are essential for plant diversification, differentiation, and the establishment of specific characteristics. For instance, the genomic rearrangement of the opium poppy (*Papaver somniferum*) [30] led to the emergence of BIAs alkaloid metabolism. The antitussive and anticancer drug noscapine, which is a member of the phthalideisoquinoline subclass of BIAs [37], is produced by a cluster of 10 genes that code for enzymes and a P450 oxidoreductase gene fusion [38,39,40] and is in charge of the crucial gateway reaction that shifts metabolites in favor of the morphinan branch and away from the noscapine branch. Wang et al. (2018) [32] reported that polyploidy contributed to the expansion of key functional genes, e.g., vitamin C biosynthesis genes in kiwifruit (*Actinidia chinensis*).

All angiosperms sequenced so far exhibit evidence of whole-genome duplication, which causes the increase of gene families and metabolic variety across evolutionary time. These complicate the studies on plant metabolism, particularly secondary metabolism. Segmental and tandem duplications and other partial genome duplications also contribute to the enormous gene family size seen in plant genomes. Genes that have been duplicated may undergo “neofunctionalization”, “sub-functionalization”, or even “pseudogenization”, which is the division of functions into duplicates [41]. Compared to primary metabolism genes, secondary metabolism genes have the following characteristics: a propensity to be preserved after gene duplication, lineage specificity, and physical grouping within the genome [42].

## 4. Omics Tools for the Study of Secondary Metabolites

Research has focused greatly on how medicinal plants produce and manage their bioactive components. In the first two decades of the 21st century, we saw a revolution in genomic technologies, and these developments continue to impact all facets of plant biology, including plant metabolism. This is because access to the genome sequence and the large data sets associated with it, such as expression and metabolite profiles, can facilitate the identification of the biosynthetic and regulatory pathways for both distinct metabolites and broad classes of metabolites. The study of “omics” has opened new avenues for examining a specific medicinal plant’s biosynthetic process (Figure 1). Creating de novo genome and transcriptome sequences, which have proven very informative across many species for understanding plant metabolism, is one apparent use of genome sequencing [43]. Many plant species now have genome sequences that may be utilized as a starting point for secondary metabolite pathway identification due to recent advancements in assembly methods and annotation software as well as access to more powerful computing resources and genome technologies [44].

The secondary metabolites biosynthesis pathway analysis will be significantly accelerated by using bioinformatics and functional genomics approaches to screen and identify enzyme-coding genes on particular secondary biosynthesis pathways from many of the original medicinal plant species. For instance, Liu et al. (2021) [2] published high-quality chromosome-scale genome assembly, annotation, and metabolomics for *Coptis chinensis* Franch. by the integrated approach of omics tools; they discovered that the (S)-canadine synthase enzyme, which is involved in the berberine biosynthesis pathway, is encoded by the *CYP719* gene. The cytochrome P450 (*CYP728B70*) gene has been shown in similar studies to catalyze the oxidation of a methyl to the acid moiety of dehydroabietic acid in the triptolide biosynthesis [3]. According to research on the *Salvia miltiorrhiza* tanshinones biosynthetic route, *CYP71D373* and *CYP71D375* catalyze the hydroxylation at carbon-16 (C16) and the 14,16-ether (hetero)cyclization to create the D-ring, whereas *CYP71D411* catalyzes the upstream hydroxylation at C20 [45]. In addition, camptothecin biosynthesis was reported in *Camptotheca acuminata* by Kang et al. (2021) [46]. It is crucial to make the most of genetic data to determine the biosynthetic pathways through which medicinal plants create their active components. Candidate genes in these pathways may then be employed in synthetic biology for heterologous bioproduction. Table 1 displays the key related genes for various recently identified metabolites.

### 4.1. Transcriptomics

Transcriptome sequencing is currently a standard experimental technique for identifying new transcripts, measuring the level of gene expression, and discovering novel genes. Studies on plants’ genetics and functional genomics have been transformed by transcriptome sequencing, especially for non-model species lacking sequenced genomes [67]. The transcriptome analysis of commercially significant plants has advanced significantly over the last ten years because of developments that overcame medicinal plant genome sequencing difficulties. The prospective method for illuminating biological processes, such as the manufacture of bioactive substances, growth and development, and the genetic variety of medicinal plants, is transcriptome analysis [68,69,70]. Fast-track breeding of medicinal plants is made possible by identifying essential genes and genetic markers associated with manufacturing secondary metabolites [71]. Transcriptomics, the comprehensive study of transcription, genomics, and proteomics, has unquestionably aided in developing a systems biology approach, building on the substantial advancements in high-throughput technology (microarrays, automated sequencing, and mass spectrometry).

Nevertheless, data mining has been streamlined, speeding up the discovery process thanks to practical computational tools (intelligent data networks, query, retrieval, analysis, and visualization tools). For microarray and RNA-seq investigations, transcriptomics approaches are highly parallel and need extensive processing to obtain valuable data. Each spot on a microarray chip represents a distinct oligonucleotide probe, and fluorescence intensity directly measures the abundance of a particular sequence. Spots are sequenced one nucleotide at a time in a high-throughput sequencing flow cell, with the color at each round representing the next nucleotide in the sequence. Other iterations of similar methods use more or fewer color channels [72]. RNA-seq studies provide a significant amount of raw sequence reads, which must be processed to produce useable data. Depending on the experimental design and aims, various bioinformatics software packages are often needed for data analysis. The process has four steps: quality control, alignment, quantification, and differential expression [73]. A command-line interface is used to execute the most well-liked RNA-seq tools in a Unix environment or inside the R/Bioconductor statistical environment [74].

The express sequence tag (EST) technique, a standard transcriptome analysis method, has given a quick and affordable alternative path toward the identification of genes encoding particular enzymes and (or) regulatory factors involved in secondary biosynthesis pathways [75]. At present, the transcriptomes of a large number of medicinal plants, such as *C. chinensis* [2], opium poppy [30], *Artemisia annua* [47], *Camptotheca acuminata* [46], *Salvia miltiorrhiza* [45,76,77], and *Stevia rebaudiana* Bertoni [78], were published online. These transcriptome data identified genes involved in the manufacture of benzylisoquinoline alkaloids, glycyrhizin, artemisinin, camptothecin, saikosaponins, taxol, tanshinone, salvianolic acid, and steviol glycoside. Table 2 lists the genes for secondary metabolite production that were found after the genomes of several medicinal plants were assembled, and transcriptomics data were merged. Genes from *Trachyspermum ammi* that are involved in thymol oil synthesis [79] and eight putative diterpene synthases (*diTPS*) of *Tripterygium wilfordii* [80] that are involved in triptolide biosynthesis have been identified by comparative transcriptome analysis. Chang et al. (2022) [81] used RNA-seq-based data from five tissues to assess the expression of the 104 moringa genes identified in their study as putatively involved in glucosinolate biosynthesis. Of the 104 genes, 84 and 88 were expressed in seeds and leaves, respectively, where glucosinolate biosynthesis is more abundant.

Co-expression of genes is common among those involved in producing secondary metabolites. Based on the transcriptome data of many samples, co-expression analysis is used to screen genes engaged in a particular pathway and reduce the number of candidate genes. Transcriptome analysis of the tanshinone production in *S. miltiorrhiza* revealed the co-expression of six cytochrome P450 genes with *SmCPS* and *SmKSL* [103]. The comparative transcriptome studies have also been used to screen the potential genes for co-expression in the production of lignans and phenylethanoid glycosides in the three species of Forsythia, *F. suspensa*, *F. viridissima*, and *F. koreana* [104]. The chromosomal-scale genome assembly made it possible to examine medicinal plants’ high contiguity and coverage transcriptome and greatly aided in identifying genes involved in manufacturing secondary compounds.

### 4.2. Metabolomics

The emergence of omics-based approaches such as genomics and metabolomics has highlighted the potential to identify new plant secondary metabolites and give extensive data on the regulatory networks of genes synthesizing the secondary metabolites. An effective platform for detecting and identifying known and unidentified secondary metabolites and enhancing secondary metabolite synthesis is provided by combining metabolomics with gene editing methods such as CRISPR-Cas9 [105]. Modern plant metabolomics, a more advanced plant metabolism, has sped up gene discovery and clarified several natural product biosynthesis pathways [106]. Metabolomics plays two roles in these studies: (1) identifying the target phytochemical’s spatial and temporal distribution as influenced by plant development and environmental cues and (2) identifying related compounds that may be thought of as alternative products of promiscuous enzymes or as intermediates in the biosynthesis of the target phytochemical. Such information may open doors to discovering biosynthetic pathways when combined with the generalizable principles of organic synthetic chemistry. Therefore, integrating metabolomics with genome-based functional characterizations of gene products enables the fast identification of new biosynthetic routes to specialized metabolites.

Spectrophotometric techniques that were compared to a certain pure standard can fall short of the actual ones. A newly created LC-MS-based, extensively targeted metabolomics approach is used to ensure the differences in total contents and composition [107,108]. The ultra-high resolution capacity of FT-ICR-MS, which also permits the inference of the metabolite annotations following database searches, is used to identify the chemical composition of the metabolomic peaks found in the LC-MS study [109]. Metabolites have been subjected to hierarchical clustering analysis (HCA), principle component analysis (PCA), and partial least squares–discriminant analysis (PLS-DA) to investigate metabolite species-specific accumulation [110]. Extensive LC-MS metabolic profiling of different tissues revealed at least 54 flavonoid compounds in A. thaliana, including 35 flavonols, 11 anthocyanins, and 8 proanthocyanidins [111]. Furthermore, two P450s, i.e., *CYP88D6* [112] and *CYP72A15486*, catalyze the oxidation of β-amyrin to glycyrrhetinic acid in *Glycyrrhiza uralensis* (licorice), and were discovered by an elegant analysis integrating expressed sequence tag studies, gene expression analyses, and metabolic profiling. In addition, metabolomics has been used to identify the glycosyltransferases that catalyze the following “tailoring processes” of triterpene saponins. In *M. truncatula*, an assortment of co-expressed genes led to the discovery of the enzyme *UGT73F3*, which glucosylates hederagenin at the C-28 position [113]. Tryptophan decarboxylase and secologanin synthase, the genes encoding the enzymes that catalyze the first committed reactions in the biosynthetic pathways, are suppressed in RNA interference lines used in metabolomics research to identify candidates for intermediary metabolites in the biosynthetic pathway [114]. A wealth of knowledge on the biological conditions of medicinal plants is provided through the integration of omics, particularly metabolomic profiling and transcriptomics. The advancement of molecular biology technology may open the door for new omics methods. Transcriptomics, genomics, and proteomics may be utilized in conjunction with the integration of transcriptomics and metabolomics to aid in the analysis of molecular biology and the biosynthetic pathway in plant secondary metabolism studies.

### 4.3. Proteomics

There is a logical need to understand how this genetic information is fully expressed, which has given rise to the fields of transcriptomics, metabolomics, and proteomics. The development of advanced genomics capabilities is associated with the ability to comprehensively and quickly determine and assemble plant genomes. Identification of proteins involved in biological processes is made possible by proteomics, which is the systematic examination of (differentially) expressed proteins. Studies of the proteins involved in the metabolic pathways that result in secondary metabolites have previously been conducted using proteomics in plant sciences [115]. The time-consuming chemistry of the intermediates in the present method of protein identification, enzyme separation, and characterization makes it challenging to identify regulatory or transport proteins. Since the proteomic technique allows for the discovery of regulatory and transport proteins in addition to enzymes, it may be quicker and more thorough. One may identify highly conserved proteins by comparing a protein’s sequence to those of *Arabidopsis thaliana* and other plant species [116]. Dedicated EST databases have been developed and used to identify proteins [117]. Model organisms have a few SM, making sequence annotation a problematic task. Early on, following the publication of the grapevine [118,119] genome sequence, functionally annotating proteomic research has shown to be an effective technique. This method involves retrieving gene ontology (GO) words from the most closely related protein using a BLAST similarity search using the BLAST2GO tool [120]. A minority of annotations are carefully selected and based on experimental evidence, albeit caution is advised since many of them are excessively generic.

However, the structural and quantitative data obtained using various proteomic approaches remain crucial for locating the target metabolites and the acting metabolic enzymes co-localizing in the cell. It can also be used to characterize protein complexes and, to a lesser extent, to analyze protein–protein interactions. As a result of everything mentioned above, proteogenomics, a cutting-edge technique for genome annotation, was created [121]. Proteogenomics seeks to uncover variant or unidentified proteins in bottom-up proteomics by searching transcriptome or genome-derived bespoke protein databases. However, actual studies show that the peptide identifications produced by these massive proteogenomic databases are less sensitive. Several methods have been suggested, such as creating limited transcriptome-informed protein databases, which only include proteins whose transcripts have been found in the sample-matched transcriptome. It was discovered that they improved peptide identification sensitivity [122]. The microsomal fraction of *Nicotiana tabacum* trichomes underwent extensive fractionation, peptide identification by LC MALDI MS/MS, and sequence analysis for transmembrane span prediction, which resulted in the identification of 165 membrane proteins, of which 39 were putative transporters, and eight were of the ABC type [123]. A number of proteins, including alcohol dehydrogenases, terpene cyclases, cytochrome P450-dependent monooxygenases, and glycosyl- and methyl-transferases, has been proposed as candidates to be involved in other unidentified steps of this pathway in addition to being identified by analogy with other plant species of already characterized proteins that act in well-defined steps of the TIA biosynthesis pathway [117]. The capacity to effectively access the SM proteome has depended on the availability of species-specific nucleotide databases and the technological technique, highlighting the potency of combining deep mRNA and protein sequencing to access plant-specific SM. After a 2-DE-based process in opium poppy (*Papaver somniferum*) cell cultures, only one enzyme implicated in the production of BIAs was discovered [124]. In contrast, the identification of 1004 proteins, which comprised almost all of the pathway’s enzymes, was made possible by the earlier construction of a particular EST database from deep transcriptome analysis, followed by an LC-MS/MS analysis of 1DE-fractionated complete protein extracts [125]. However, unlike the study of DNA, proteins provide some particular difficulties. For instance, proteins do not have a PCR equivalent, making it challenging to analyze low-quantity proteins. Additionally, natural protein conformations must be preserved to obtain significant findings from investigations of protein interactions.

Additionally, discovery investigations in plant SM have been carried out using differential proteomics techniques that outperform single staining 2-DE. These include differential in gel electrophoresis (DIGE) as a top-down protein-centric approach [126] and isobaric tags for relative and absolute quantitation (iTRAQ) [127] and label-free [128] as bottom-up, peptide-centric approaches. Among the 172 unregulated proteins discovered by DIGE, three TIA biosynthesis enzymes were found in *Catharanthus roseus* cell cultures along with five more enzymes that may be putatively implicated in the process [117]. When DIGE and iTRAQ data were compared, they revealed significant quantitative correlations between frequently recognized proteins although iTRAQ excels in proteome coverage. However, the slight overlap between the two proteome studies highlights the need to use several strategies better to comprehend the target process under investigation [127]. Even though protein analysis technology is developing quickly, it is still impossible to analyze proteins on par with nucleic acid research. Additionally, these methods must be error-tolerant of accounting for sequencing mistakes, polymorphisms, and conservative substitutions to access unannotated DNA libraries from other species. Before protein analysis on a broad scale (such as mapping the proteome of medicinal plants) becomes a reality, new methods must be developed.

### 4.4. Multi-Omics Approaches

New metabolite discoveries and a new method of understanding biosynthetic pathways have been made possible by the advent of functional genomics, including transcriptomics, metabolomics, and proteomics [129]. The multi-omics study may significantly elucidate the diversity, location, and evolution of BGCs [84]. Genes synthesizing specific metabolites may often be identified by the association between their expression and the accumulation of related metabolites [130]. To demonstrate variations in metabolites and gene pathway enrichment concurrently, differentially expressed genes (DEGs) and differentially expressed metabolites (DEMs) are mapped to KEGG pathways. Each group’s DEGs and DEMs undergo correlation analysis, and genes’ and metabolites’ Pearson correlation coefficients are computed. Canonical correlation analysis is utilized to represent the overall correlation between the transcriptome and metabolome [131]. The O2PLS model is built using all DEGs and DEMs, and the variables with the highest correlation and weight across datasets are chosen based on the load diagram, whereas significant variables that impact another set are filtered out [132]. For instance, the relative abundance of transcripts encoding the alkaloid biosynthetic enzymes correlates with the induction of BIAs accumulation in *Papaver somniferum* [133]. Zhou et al. (2021) [83] combined metabolomics and transcriptomics analyses to explore the bioactive constituent biosynthesis in the leaves, stem, and root of *Perilla frutescens* (L.) Britt. Transcriptome sequencing profiles revealed that terpenoids, flavonoids, and phenylpropanoids biosynthetic genes are regulated differently, with most of these genes highly expressed in leaves. Transcriptomics, proteomics, and metabolomics were used by Cristina et al. (2018) [82] to study the biosynthesis of secondary metabolites, carbohydrates, energy, amino acid, lipid, and nucleotide metabolism in the orphan species *Quercus ilex*.

Moreover, a comprehensive multi-omics analysis report by Liu et al. (2019) [76] employed nuclear magnetic resonance (NMR)-based metabolomics and transcriptomics techniques to study the biosynthesis pathway of salvianolic acids in *Salvia miltiorrhiza* Bunge. The recent studies on discovering secondary metabolite biosynthetic genes in medicinal plants by combining different omics data analyses are summarized in Table 2. A group of FAD2-related enzymes has been found by comparing the transcriptomes and metabolomes (in particular, fatty acids and lipids) of growing seeds that accumulate “unusual” fatty acids [134] that are responsible for the generation of hydroxy fatty acids, epoxy-fatty acids, conjugated fatty acids, and acetylenic fatty acids. A relatively complete biosynthetic pathway for the three *O*-methylated flavones, namely hispidulin, jaceosidin, and eupatilin, was conjected based on synergistic genome sequencing analysis transcriptomics and metabolomics [135]. Transcriptomics, phenolic metabolomics, and functional gene analyses were used to characterize an aggregation gene cluster associated with the biosynthesis of hydrolyzable tannins (HTs) [65]. Hence, comparing transcriptomics and metabolomics data enables the precise annotation of a wide variety of genes in specialized metabolism.

### 4.5. Other Tools and Platforms for Medicinal Plant Functional Genomics

#### 4.5.1. Metabolite-Based, Genome-Wide Association Study (mGWAS)

To analyze plant metabolism’s genetic and biochemical underpinnings, metabolite-based, genome-wide association studies (mGWAS) have become a potent alternative forward genetics technique [136]. mGWAS in plants gets extra benefits from both huge diversity [130] and high inheritability of most metabolites [137,138] (Figure 2). It is feasible to develop testable hypotheses that can be confirmed in subsequent experiments since putative associations’ biological and functional justifications in mGWAS are corroborated. Candidate genes can be discovered by searching for a protein or protein cluster that is biochemically related to the metabolic trait associated with it and encoded at the associated loci by performing cluster analysis of candidate genes concerning homologous genes with known functions and by cross-referencing with linkage mapping data. Rapid discovery of new genes and potential networks of plant metabolism is made possible by integrating GWAS findings with other types of genome-scale data, such as transcript profiling or proteomics datasets [137,138,139]. mGWAS was initially applied in the model species *A. thaliana* [140], then successfully performed and extended in some crops [137,141]. Within structured mapping populations, the interactions between genotypes, environment, and development have been shown to affect the accumulation of secondary metabolites. A considerable bias was found in mGWAS on the naturally occurring ring variation of GSL accumulation in *Arabidopsis* toward identifying several causative genes for the GSL phenotypes in the two distinct tissues and during various developmental phases [142]. It implies that the genetic regulation of the natural variation of GSLs is spatially and temporally regulated.

Interestingly, different genetic control over metabolism was also imposed at the subspecies level [137]. Discovering the purposes of each gene in plant genomes is the goal of functional genomics. mGWAS may be used in functional genomics in addition to illuminating the genetic structure of plant metabolism. For example, mGWAS has identified 36 candidate genes encoding metabolites of physiological and dietary value using extensive knowledge of several plant metabolic pathways [137]. Two spermidine hydroxycinnamoyl transferases were responsible for the normal variation of levels of spermidine conjugates in rice, according to a detailed analysis of spatiotemporal accumulation and natural variation of phenolamides in rice, followed by mGWAS [143]. However, regarding the use of mGWAS in medicinal plants, there is no history. The reconstruction of biosynthetic pathways is made more accessible by understanding natural variation at the metabolic level using mGWAS, which is advantageous for biomedical research and the metabolic engineering of desired plant secondary metabolites [137,144,145].

#### 4.5.2. Metabolic Quantitative Trait Loci (mQTL)

A practical method for identifying the genetic underpinnings of complex features in organisms such as medicinal plants is quantitative trait loci (QTL) mapping [146]. The self-incompatibility of medicinal plants, which results in limited populations and poor seed production, makes applying this strategy difficult. Despite this, more attention is being paid to the QTL mapping of numerous agronomic parameters of medical plants, notably yield, and secondary metabolites given the significance of medicinal plants in worldwide consumption [147,148,149,150,151]. Quantitative trait loci (QTL) analysis was employed in recent research to identify and pinpoint the locations of genomic areas that influence variance for quantitative traits since many aspects contributing to the phytochemical composition of medicinal plants are quantitative [152]. For example, Zhang et al. (2020) [153] have reported the application of QTL in ginsenoside biosynthesis in ginseng (*Panax ginseng* C.A. Meyer). They discovered that when expressed in the medicinal plant, most genes regulating quantitative traits were noticeably more likely to be spliced into numerous transcripts. The application of QTL mapping is also reported in several crops [127,128,129].

To understand the nature of inheritance of the amount and composition of these plant metabolites, metabolic quantitative trait loci (mQTL) investigations for targeted and non-targeted metabolites of therapeutic significance have also been carried out in numerous plant species [152]. In mQTL, MetaNetwork [154] software is used to map the metabolite variation. For example, 25 catechin-related QTLs were found using 2-year catechins data from 183 individuals crossbred from two tea plant kinds with different catechin compositions. Nine of these QTLs were confirmed throughout time and grouped in the chromosomal areas LG03 and LG11, pointing to a possible tandem duplication origin for the catechin genes in tea plants [149]. Similarly, 148 individuals from a pseudo-testcross population were used to determine theobromine and caffeine levels using 10 QTLs [148]. Recent research on QTL mapping discovered multiple new flavonoid-related QTLs due to the growth of the mapping populations [151]. To correctly identify QTLs in medicinal plants, more research will be required to expand the mapping populations and enhance the density of genetic linkage maps. Metabolites, on the other hand, are mechanistically more distant from the genome than mRNAs, leading to a crucial qualitative difference between metabolite QTL (mQTL) and eQTL studies. An advantageous feature of eQTL investigations is the ability to map mRNA to genes, which enables the search for a cis eQTL of each mRNA to be concentrated on a very narrow, gene-centered region. The connection between metabolite variation and genome-wide genetic variation is investigated in mQTL studies. Since there are many tests run, the effect sizes need to be much greater to be statistically significant. As a result, mQTLs are often more challenging to find than eQTLs with equal effect sizes in addition to perhaps being rarer.

#### 4.5.3. Bulked-Segregant Analysis (BSA)

The recently developed BSA technique chooses and pools data from individuals with extreme phenotypes from biparental segregation populations, using the practical discovery of genes and alleles governing complex features via statistical genomic and phenotypic investigations. Bulked segregant analysis sequencing (BSA-seq) may be used to find the allele frequency (AF) of single-nucleotide polymorphisms by generating a DNA pool of offspring with extreme characteristics and utilizing molecular markers to analyze co-segregation analysis between markers and traits in the two pools (SNPs). Therefore, BSA-seq allows the quick identification of essential genes linked with quantitative features or genes associated with key qualitative qualities regulated by a single gene [155]. The SNP-index technique was used to conduct SNP annotation, identify the effects (synonymous and nonsynonymous mutations) of small indels [156] in the genome, and compare the variations in allele frequency across bulked pools [157,158]. The SNP index was formerly referred to as the number of short reads with SNPs that vary from the reference genome [159]. For instance, based on samples from the two tails of an F1 population that segregates with catechin concentration, the BSA technique, in conjunction with RNA sequencing, has aided the finding of a flavonoid three 5-hydroxylase (F3′5′H) genes in tea plants [160]. Studies conducted on the tomato and *Arabidopsis* have shown that the content of plant metabolites exhibits transgressive segregation, thus offering an opportunity to develop improved cultivars [161,162]. Traditionally, forward and reverse genetic screening, positional cloning, and quantitative trait loci mapping have been used to find causative genes within a metabolic pathway. High-throughput sequencing technologies may speed up and improve the efficiency of these traditional approaches. These sequencing technologies are not only capable of quickly detecting sequence variations (SNPs and insertions/deletions) across accessions; it can also be utilized in methods such as whole-genome resequencing in bulk segregant [163] analysis and *k-mer*-based [164] techniques to identify causative mutations. A single laboratory may currently employ sequencing techniques in combination with genetics methodologies for gene discovery because of the cheap cost, high coverage, accuracy, and accessible bioinformatics pipelines. For instance, having access to the genome sequences for several accessions enables association research to use linkage mapping or a genome-wide association technique to connect, for example, metabolite with genotype. With access to extensive transcriptome data sets, expression quantitative trait loci (eQTLs) associated to the production, control, or transport of metabolites may be found by tying transcript levels to sequence variations.

#### 4.5.4. Weighted Gene Co-Expression Network Analysis (WGCNA)

The relationship between phenotype, including metabolites, patterns, and gene expression profiles, is a correlative method for annotating gene function. Access to genome sequencing enables the quick and affordable collection of expression abundance data. Coordinated expression studies (co-expression) may uncover genes within the same regulatory network from an expression atlas, in which various tissues and treatments are evaluated. WGCNA has effectively identified a subset of potential biosynthetic pathway genes for functional validation combined with examining the functional annotation of genes or transcripts [165]. WGCNA is a powerful approach to developing the regulatory network [166,167], and it is usually done in the R software WGCNA package [167,168]. Genes that were differently expressed in *Camptotheca acuminata* were used by Kang et al. (2021) [46] to create WGCNA. They discovered 30 clusters, including the seco-iridoid pathway genes geraniol synthase (*GS*), iridoid synthase (*IS*), 7-deoxyloganetic acid synthase (*7-DLS*), 7-deoxyloganetic acid O-glucosyltransferase (*7-DLGT*), and 7-deoxyloganic acid 7-hydroxylase (*7-DLH*), as well as other related genes, which were grouped. The WGCNA was also utilized by Tai et al. (2018) [169] to identify 35 co-expression modules in the tea plant, of which 20 modules were substantially linked to the production of catechins, theanine, and caffeine. The presence of 3 of the 20 co-expression modules, which included six genes involved in ascorbic acid biosynthesis and regeneration pathways, was shown to be positively linked with the ascorbic acid content of the guava (*Psidium guajava*), according to research on WGCNA [170]. According to WGCNA results in *indica-japonica* rice hybrids, ribosomal protein production and stress response were tightly related to the bisque4 and darkorange2 modules for types 1 and 3 of the core metabolites. Eight hub genes, namely *rpL32 8.1*, *novel.907*, *RPS15a*, and *RPL13a.4* in bisque4 as well as *OsSAP5*, *OsHSP20*, *PRX12*, and *DUF3741* in darkorange2, were discovered [171]. At various phases of the *C. panzhihuaensis* ovule and seed’s development, 11 co-expression modules were discovered by Liu et al. (2022) [172]. In summary, it enables the definition of modules (clusters), intra-modular hubs, and network nodes regarding their participation in modules as well as the analysis of the interactions between co-expression modules and the comparison of the network topologies of other networks (differential network analysis).

## 5. Conclusions and Prospects

Since the establishment of the human genome project (HGP), there has been a significant development regarding the high-throughput sequencing technologies, and accurate genome assembly has enabled the sequence of the medicinal plant genome at the chromosomal level in a short time and at low cost. Functional genomics (proteomics, transcriptomics, and metabolomics) paved many ways to identify key related genes and enzymes for studying secondary metabolite biosynthesis pathways. Furthermore, the combination of multi-omics provides a complete picture of medicinal plants, and their metabolism could identify novel genes and metabolites. Yet, there is not enough information regarding the application of mQTL, BSA, WGCNA, and mGWAS in medicinal plants, and our understanding of them needs to be strengthened.

However, high-throughput sequencing technologies and functional genomics reduced the cost and the complexity of DNA sequencing; many medicinal plants’ genomes were sequenced and assembled, and shortly, this number will be increased; the multi-omics and bioinformatics approaches can be the best tools for the study of secondary metabolism. The promotion of medicinal plant genome projects contributes to the biosynthesis and screening of plant substrates and plant-based drugs and prompts the research efficiency of traditional medicine.

## Figures and Tables

**Figure 1 ijms-23-15932-f001:**
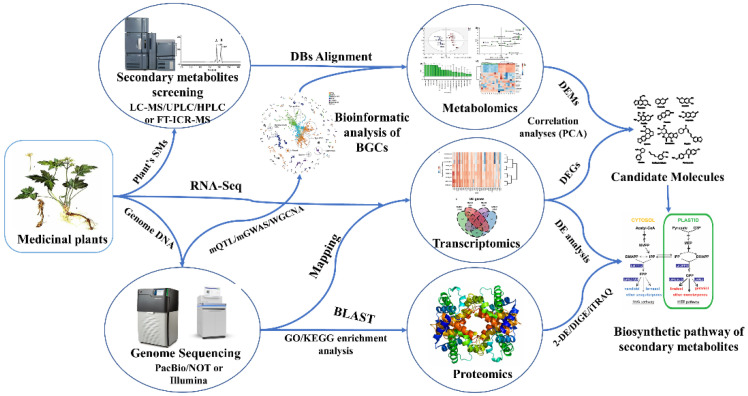
The multi-omics strategy and its applications. Numerous properties of a particular molecular type may be simultaneously measured in biological samples using various omics approaches. Data integration, including bioinformatic tools, pipelines, and databases, is crucial in identifying secondary metabolite biosynthesis pathways.

**Figure 2 ijms-23-15932-f002:**
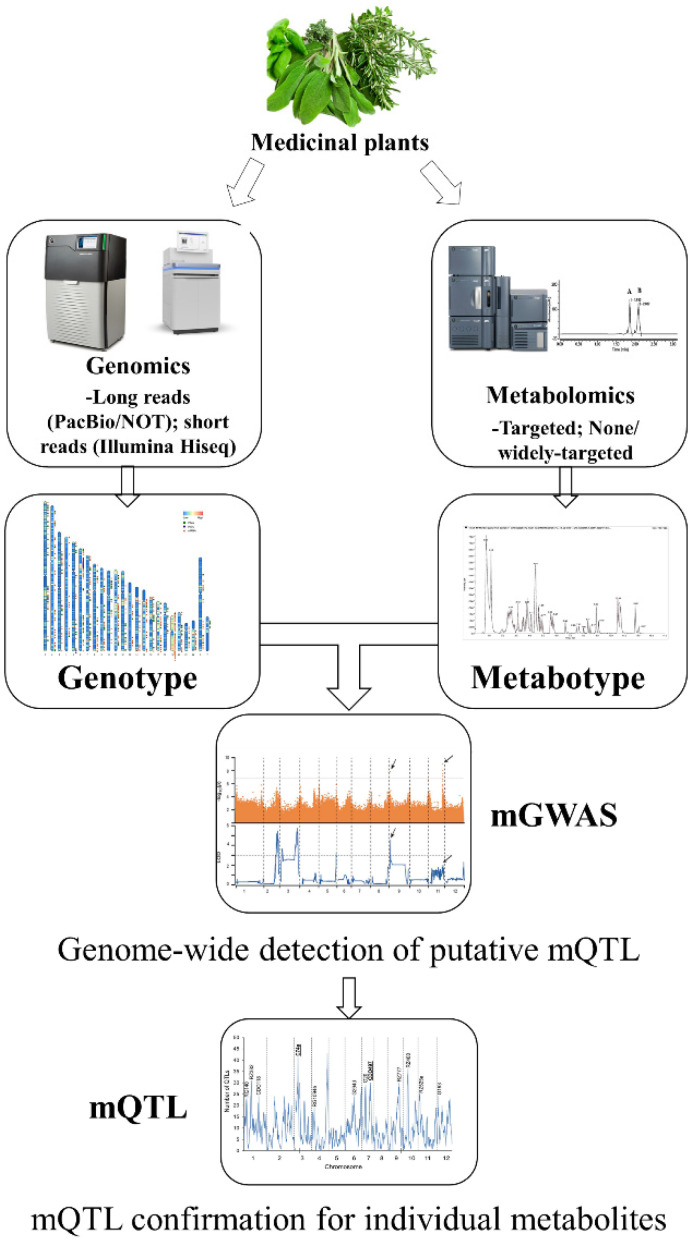
Flowchart of mGWAS. Combining the relevant data on the chemical structures and pathway architectures of associated/linked metabolites with the reported ortholog functions on metabolite analogs, candidate genes responsible for the mGWAS/mQTL assays loci are discovered.

**Table 1 ijms-23-15932-t001:** Third-generation sequencing technology for studying secondary metabolites biosynthesis pathway.

Medicinal Plants	Sequencing Technologies	Identified Genes	Biosynthetic Pathways	Reference
*Coptis chinensis* Franch.	PacBio, O.N.T., and Hi-C	Cytochrome P450 (*CYP719A*)	BIAs	[2]
*Tripterygium* *wilfordii*	PacBio, 10X genomics, and Hi-C	Cytochrome P450 (*CYP728B70*)	Triptolide	[3]
*Artemisia annua*	PacBio, Roche 454, and Illumina	*ADS*, *CYP71AV1*, and *FPS*	Artemisinin	[47]
*Piper nigrum*	PacBio, 10X Genomics, O.N.T., and HI-C	*GTF*, *CYP*, and *HCT*	Piperine	[48]
*Allium sativum*	PacBio, Illumina, and 10X Genomics	*AsGSH1b*, *AsGSH2*, *AsPCS1*, *AsFMO1*, and *AsGGT2*	Allicin	[49]
*Magnolia biondii* Pamp.	PacBio, 10X genomics, and Hi-C	*TPS* gene family	Terpenoids	[50]
*Camptotheca acuminata*	PacBio, Illumina, and Hi-C	*LAMT* and *SLAS*	Camptothecin	[46]
*Platycodon grandiflorus*	Illumina HiSeq 2500	*CYP716* and *bASs*	Platycoside	[51]
*Ophiorrhiza pumila*	ONT, Illumina, and Hi-C	*ASO/PAS*, *PNAE*, *PR*, *RH11H*, *SBE*, *SGD*, *T19AT*, and *THAS*	Camptothecin	[7]
*Strobilanthes cusia*	BioNano, Illumina, and Hi-C	*CYP*, *FMO*, *UGT*, and *BGL* gene family	Indigo	[52]
*Scutellaria baicalensis*	PacBio, Illumina, and Hi-C	*CHS-2*, *FNSII-2*, *F8H*, and *PFOMT5*	Wogonin	[53]
*Scutellaria baicalensis* and *Scutellaria barbata*	PacBio, ONT, and Hi-C	*PAL*, *4CL*, *CHS*, *F6H*, and *F8H*	Flavonoid	[17]
*Chiococca alba*	Illumina HiSeq 4000 and 10X Genomics	*TPSs* gene family	Unusual terpenoids	[54]
*Acer truncatum*	PacBio, Illumina, 10X Genomics, and Hi-C	*KCS*	Nervonic acid	[55]
*Opium poppy*	PacBio, Illumina, 10X Genomics, and ONT	*CYP 450* and *STORR*	Morphinan	[30]
*Senna tora*	PacBio, Illumina, and Hi-C	*CHS-L*	Anthraquinone	[56]
*Hypericum perforatum*	PacBio, Illumina, and 10X Genomics	*HpASMT1*, *HpASMT2*	Melatonin	[57]
*Vernicia fordii* Hemsl.	Illumina HiSeq 2000	*FAD2*, *FADX*	Triacylglycerol	[58]
*Andrographis paniculata*	PacBio, Illumina, and Hi-C	*diTPSs*, *CYP450*, *2OGDs*, and *UGT*	Diterpenoid neoandrographolide	[59]
*Eucommia ulmoides*	PacBio, O.N.T., and Illumina	*FPSs*	Polyisoprene	[60]
*Gardenia jasminoides*	ONT and Hi-C	*GjCCD4a*, *ALDH*, and *UGT*	caffeine and crocin	[61]
*Salvia bowleyana*	PacBio, Illumina, and Hi-C	*SbPAL1*	Salvianolic acid B	[62]
*Chimonanthus salicifolius*	Illumina, PacBio, 10X Genomics, and Hi-C	*CHS* and *FLS*	Flavonoid	[63]
*Strobilanthes cusia*	PacBio, Illumina, and Hi-C	*UGT*, *IGPS*, *CYP450*, *EPSPS*, and *CS*	Indole alkaloids	[64]
*Salvia miltiorrhiza*	PacBio, Illumina, and Hi-C	*CYP71D373*, *CYP71D375*, and *CYP71D411*	Tanshinones	[45]
*Rubus chingii* Hu	ONT, Illumina, and Hi-C	*C.X.E.*, *U.G.T.*, and *SCPL*	Hydrolyzable tannin	[65]
*Tripterygium* *wilfordii*	ONT, Illumina, and Hi-C	*TwCYP712K1* and *TwCYP712K2*	Celastrol	[66]
*Papaver*	ONT, Illumina, and Hi-C	*BIA gene*	Morphinan and noscapine	[66]

**Table 2 ijms-23-15932-t002:** Multi-omics approaches to study the biosynthesis pathway of secondary metabolites.

Species	Omics Techniques	Metabolites	References
*Salvia miltiorrhiza* Bunge.	NMR-based metabolomics, transcriptomics	Salvianolic acids	[76]
*Quercus ilex*	Metabolomics, proteomics, transcriptomics	62 metabolites	[82]
*Perilla frutescens*	Metabolomics, transcriptomics	Terpenoids, flavonoids, and phenylpropanoid	[83]
*Rosa roxburghii* Tratt.	Metabolomics, transcriptomics	Amino acid, phenylpropanoid, and flavonoid	[84]
*Astragalus membranaceus* Bge.	Metabolomics, transcriptomics	Phenylpropanoid, flavonoid, and isoflavonoid	[85]
*Chenopodium quinoa* Willd.	Transcriptomics, metabolomics	Flavonoid	[86]
*Triticum aestivum* L.	Transcriptomics, metabolomics	Phenolic	[87]
*Gardenia jasminoides* Ellis.	Transcriptomics, metabolomics	Iridoid and crocin	[88]
*Flammulina velutipes*	Transcriptomics, metabolomics	Ergosterol	[89]
*Perilla frutescens* L.	Transcriptomics, metabolomics	Flavonoid	[90]
*Euphorbia lathyris* L.	Transcriptomics, metabolomics	Ingenol	[91]
*Triticum aestivum* L.	Transcriptomics, metabolomics	Anthocyanin	[92]
*Setaria italica* L.	Transcriptomics, metabolomics	Phenylpropanoid, flavonoid, and lignin	[93]
*Salvia miltiorrhiza*	Transcriptomics, metabolomics	Tanshinone	[77]
*Acer mandshuricum*	Transcriptomics, metabolomics	Anthocyanin	[94]
*Corydalis yanhusuo*	Transcriptomics, metabolomics	Benzylisoquinoline alkaloid	[95]
*Zanthoxylum schinifolium* Sieb.	Transcriptomics, metabolomics	Phenylpropanoid, flavonoid, flavone, and flavonol	[96]
*Leptobryum pyriforme*	Transcriptomics, metabolomics	Flavonoid	[97]
*Angelica sinensis*	Transcriptomics, metabolomics	Phenylpropanoid	[98]
*Dendrobium huoshanense*	Transcriptomics, metabolomics	Flavonoid	[99]
*Solanum lycopersicum L.*	Transcriptomics, metabolomics	Phenolamide	[100]
*Dendrobium sinense*	Transcriptomics, metabolomics	Purine and phenylpropanoid	[101]
*Nicotiana tabacum L.*	Proteomic, metabolomic	Aroma precursors	[102]

## Data Availability

Not applicable.

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
