# Peer review of "The Current Developments in Medicinal Plant Genomics Enabled the Diversification of Secondary Metabolites’ Biosynthesis"

_ijms, 2022, doi:10.3390/ijms232415932_

Round 1

Reviewer 1 Report (Previous Reviewer 2)

I do not have additional comments.

Author Response

We thank the reviewer for reviewing our manuscript.

Reviewer 2 Report (New Reviewer)

Presented review describes Omics aspects of medicinal plants with regards to secondary metabolites. The following issues need to be resolved for the review paper.

- The explanations like "the NGS technology transformed Omics sector," are repetitive in many paragraphs. These parts should be shorten and focus on the detailed techniques, outputs, and notable findings. 

- "4. Omics tools" part should include detailed analytical methods that successfully extract informatic genetic factors contributing secondary metabolite contents. Currently, readers are just given brief explanations of each omics and a few cases.

Author Response

General Comments: Presented review describes Omics aspects of medicinal plants with regards to secondary metabolites. The following issues need to be resolved for the review paper.

Response: We sincerely thank the reviewer for their constructive and valuable comments, which greatly help to revise the manuscript. Accordingly, the revised manuscript has been improved with new information and interpretations.

Point 1: The explanations like "the NGS technology transformed Omics sector," are repetitive in many paragraphs. These parts should be shorten and focus on the detailed techniques, outputs, and notable findings.

Response 1: Thank you! We found your comments extremely helpful. We have revised the entire manuscript and have removed the repetitive words. In addition, we have added information related to the specific NGS technologies and their application in secondary metabolites biosynthesis pathways analysis.

Point 2: 4. Omics tools" part should include detailed analytical methods that successfully extract informatic genetic factors contributing secondary metabolite contents. Currently, readers are just given brief explanations of each omics and a few cases.

Response 2: We appreciate the reviewer's feedback. We thank the reviewer for their constructive and invaluable comments and suggestions. We have added the analytical methods for the omics tools in section 4. It is better to mention some omics tools like transcriptomics, metabolomics, and the combination of these two omics (multi-omics) analytical methods consisting of many analyses. Adding the details of these methods is out of this manuscript range. We have added the exact methodologies used in these omics tools briefly. Some other analytical methods like WGCNA and mGWAS are usually done in R and Python software and have special packages and scripts. The algorithm of these tools is mentioned and cited in the manuscript. The revisions are highlighted in blue in the manuscript.

Round 2

Reviewer 2 Report (New Reviewer)

The authors have well addressed the concerns.

This manuscript is a resubmission of an earlier submission. The following is a list of the peer review reports and author responses from that submission.

Round 1

Reviewer 1 Report

The present study mainly summarized the basic information on medicinal plant genomes, including sequencing technology, genome size, sequencing coverage, and references, and also presented the reported secondary metabolic biosynthesis pathway revealed by third-generation sequencing technology. This review fails to delve into these genomic data or case studies to offer new insights. Therefore, I think this paper is not suitable for publication in this journal.

1. It is not necessary to describe the revolution of genome sequencing technologies, which is common knowledge to professionals and is not the focus of this article. In addition, there is too much basic knowledge about sequencing technology in this article, such as Bionano optical atlas and Hi-C.

2. The authors should pay special attention to grammatical and formatting errors. Many mistakes can be seen. Line 14-15, “secondary metabolites biosynthesis is very low”, “raped”, “6. multi-omics approaches to study”

3. Figure 2 was not cited in the main text. Besides, it is suggested to use an example from your own research rather than a cartoon to present the process of the comparative genomics analysis, you may use the tool Clinker.

4. The technical road for the identification of biosynthetic pathways was confusing, the figure and explanation should be reorganized.

Author Response

General Comments: The present study mainly summarized the basic information on medicinal plant genomes, including sequencing technology, genome size, sequencing coverage, and references, and also presented the reported secondary metabolic biosynthesis pathway revealed by third-generation sequencing technology. This review fails to delve into these genomic data or case studies to offer new insights. Therefore, I think this paper is not suitable for publication in this journal. It is not necessary to describe the revolution of genome sequencing technologies, which is common knowledge to professionals and is not the focus of this article. In addition, there is too much basic knowledge about sequencing technology in this article, such as Bionano optical atlas and Hi-C.

Response: We appreciate the feedback from the reviewer. As suggested by the reviewer, we have revised the manuscript systematically and highlighted them in blue. In the manuscript, the highlighted subsections' titles indicate that the entire section has been added to the manuscript. The following revisions are applied to the manuscript:

  • We have summarized the sequencing technologies in one paragraph and added them to the introduction section; the tables and figures related to this section were moved to the supplementary materials, which are named supplementary table1-3 and supplementary figure 1.
  • We have added more information and case reports to the comparative genomics and polyploidy or whole genome duplication (WGD) for the evolution of metabolites in medicinal plants.
  • We have added more sections related to the application of omics tools in medicinal plant genomics, such as Transcriptomics, metabolomics, proteomics, and some other tools in medicinal plants functional genomics like mQTL, BSA, WGCNA, and mGWAS.

Point 1: The authors should pay special attention to grammatical and formatting errors. Many mistakes can be seen. Line 14-15, "secondary metabolites biosynthesis is very low", "raped", "6. multi-omics approaches to study"

Response 1: We thank the reviewer for this comment. We have double-checked the entire manuscript and removed all the misspellings and grammatical and formatting errors.

Point 2: Figure 2 was not cited in the main text. Besides, it is suggested to use an example from your own research rather than a cartoon to present the process of the comparative genomics analysis, you may use the tool Clinker. The technical road for the identification of biosynthetic pathways was confusing, the figure and explanation should be reorganized.

Response 2: Thanks for your comments. Figures in the manuscript were not precise; thus, we have summarized them in one figure and named "figure 1". This figure shows the multi-mics tools and their application for identifying the secondary metabolites' biosynthesis pathways.

Reviewer 2 Report

Dear Authors,

This manuscript 'The current developments in medicinal plant genomics enables the diversification of secondary metabolites biosynthesis' very well thought out, however I have some comments regarding many mnistakes about technical issues. Please use different expression for 'raped' in abstract (line 15, pg1). Introduction, line 44, pg1, delete the before production, line 47, pg1 use some other term instead of 'fresh' vitality. Lines 94, 101 and 103, you do not have extra space between the sentence and reference in the brackets. Same at pg 3, line 134. After first four-five pages, in the text I do not see the numbers of lines that can be included for each page. So I can just randomly write about it: in whole mansucript mostly you do not have extra space between the word in the sentences and references in brackets, somwhere you have capital letter in the midle of sentence 'According', 'Comprehensive', Figure 1 is completely moved and standing strangely, somwhere you have latin name where you did not make italic form, somwhere you miss capital letter 'table 3'. 

My main objection to text is that I was hoping from the title of manuscript to see more about secondary metabolites in medical plants. this is missing.

Author Response

General Comments: This manuscript 'The current developments in medicinal plant genomics enables the diversification of secondary metabolites biosynthesis' very well thought out, however I have some comments regarding many mistakes about technical issues.

Response: We sincerely thank the reviewer for his/her constructive and valuable comments, which greatly help to revise the manuscript. Accordingly, the revised manuscript has been systematically improved with new information and interpretations.

Point 1: Please use different expression for 'raped' in abstract (line 15, pg1). Introduction, line 44, pg1, delete the before production, line 47, pg1 use some other term instead of 'fresh' vitality. somwhere you have capital letter in the midle of sentence 'According', 'Comprehensive', Figure 1 is completely moved and standing strangely, somwhere you have latin name where you did not make italic form, somwhere you miss capital letter 'table 3'.

Response 1: Thank you! We found your comments extremely helpful. We have revised the entire manuscript's misspelling and grammatical and formatting mistakes.

Point 2: My main objection to text is that I was hoping from the title of manuscript to see more about secondary metabolites in medical plants. this is missing.

Response 2: We appreciate the reviewer's feedback. We have revised the manuscript accordingly and added the following information related to secondary metabolism:

  • Context related to the sequencing technologies development is summarized and added as one paragraph to the introduction section, and the tables and figures related to this section were moved to supplementary materials and named supplementary table1-3 and supplementary figure 1.
  • We have added more case reports and information to the comparative genomics and polyploidy, which contribute to the evolution of secondary metabolites and duplication of key related genes in medicinal plants.
  • We have expanded the context related to the multi-omics approaches and other platforms for functional genomics, such as mQTL, BSA, WGCNA, and mGWAS.
  • We have modified the figures and summarized them in one figure named figure 1.

All the changes are highlighted in blue in the manuscript. The highlighted titles indicate that the entire section has been added to the manuscript.
